# Spinasterol, 22,23-Dihydrospinasterol and Fernenol from *Citrullus Colocynthis* L. with Aphicidal Activity against Cabbage Aphid *Brevicoryne Brassicae* L.

**DOI:** 10.3390/molecules25092184

**Published:** 2020-05-07

**Authors:** Maqsood Ahmed, Peiwen Qin, Mingshan Ji, Ran An, Hongxia Guo, Jamil Shafi

**Affiliations:** 1College of Plant Protection, Shenyang Agricultural University, Shenyang 110866, China; maqsoodahmed200@hotmail.com (M.A.); anan8u8@outlook.com (R.A.); ghx331682817@163.com (H.G.); 2Department of Agriculture, Pest Warning & Quality Control of Pesticides, Gujrat 50700, Pakistan; 3Department of Plant Pathology, University of Agriculture Faisalabad, Sub-Campus Depalpur, Okara, Faisalabad 56300, Pakistan; jamil_shafi786@yahoo.com

**Keywords:** *Citrullus colocynthis*, spinasterol, 22,23-dihydrospinasterol, fernenol, insecticidal activity, LC_50_

## Abstract

*Brevicoryne brassicae* is a problematic pest in cabbage and other field crops. Synthetic pesticides are used to control this pest, but they are injurious for human health and the environment. The present study aimed to purify and identify the active compounds from *Citrullus colocynthis* leaves with an appraisal of their efficacy against *B. brassicae*. Separation and purification were performed via different chromatographic techniques. Molecular analysis and chemical structures were recognized by mass spectrum (MS) and nuclear magnetic resonance (NMR), respectively. Moreover, in vitro and in vivo aphicidal activity was assessed using various concentrations, i.e., 6.25, 12.5, 25 and 50 µg/mL at 12, 24, 48 and 72 h exposure. The outcome shows that mass spectrum analyses of the purified compounds suggested the molecular formulae are C_30_H_50_O and C_29_H_50_O, C_29_H_48_O. The compounds were characterized as fernenol and a mixture of spinasterol, 22,23-dihydrospinasterol by ^1^H-NMR and ^13^C-NMR spectrum analysis. The toxicity results showed that the mixture of spinasterol and 22,23-dihydrospinasterol showed LC_50_ values of 32.36, 44.49 and 37.50 µg/mL by contact, residual and greenhouse assay at 72 h exposure, respectively. In contrast, fernenol recorded LC_50_ values as 47.99, 57.46 and 58.67 µg/mL, respectively. On the other hand, spinasterol, 22,23-dihydrospinasterol showed the highest mortality, i.e., 66.67%, 53.33% and 60% while, 30%, 23.33% and 25% mortality was recorded by fernenol after 72 h at 50 µg/mL by contact, residual and greenhouse assay, respectively. This study suggests that spinasterol, 22,23-dihydrospinasterol are more effective against *B. brassicae* which may be introduced as an effective and suitable substitute of synthetic chemical pesticides.

## 1. Introduction

The cabbage aphid, *Brevicoryne brassicae* L. belongs to (Hemiptera: Aphididae) is one of the key pest of vegetables and crops, commonly distributed in warm and moderate areas of the world [1,2]. Due to the heavy infestation of *B. brassicae* on cabbage, yield losses can be increased up to 70% and under-promising circumstances, it can cause complete loss of leafy vegetables of brassica [3,4]. Being an injurious pest, it negatively affects cabbage production by developing sooty mold on plants surface [2,5]. Continued feeding of this pest caused yellowing, wilting and stunting of plant growth, ultimately cause plant death and lead to economic losses [6]. Moreover, most of the species of aphid have gained resistance against many synthetic aphicidal agents [7].

Although synthetic chemicals are widely used to manage this pest but, intensive and continuous use of these chemicals has resulted in the development of pest resistance, resurgence to these chemicals and may also leave hazardous effects on humans and the environment [8]. In contrast, botanical insecticides are relatively safe to some extent and are effective substitutes of these chemicals which play a fundamental role in the field of biopesticides. However, plant derivatives play a vital role in their biologic tasks [9]. Numerous findings have specified that phenols, terpenoids and nitrogen-based constituents are imperative phytoalexins which afford a protective system to plants being attacked by other injurious insect-pests [9,10].

*Citrullus colocynthis* is an important plant from the pharmacological and pesticidal viewpoint. It belongs to the Cucurbitaceae family, mostly grows in desert areas and has attracted the consideration of researchers as a natural botanic pesticide. The insecticidal potential of this desert plant has been assessed against various insect species [11]. *Citrullus colocynthis* exerts carcinogenic, antidiabetic, antibacterial, antioxidant properties, and possesses toxic properties against several harmful insects [12,13,14,15]. Numerous active chemical compounds have been reported from *C. colocynthis*, including bitter materials (colocynthetin and colocynthin), various cucurbitacins such as A, B, C, D and E [16], other cucurbitacins like E, I, J, K and L [17], cucurbitacins glycosides [18,19], the cucurbitacins glucosides I and L [19], flavone glycosides and flavonoids [19,20]. However, the cucurbitacins tetracyclic triterpens possess extensive biologic activities. Numerous biologic compounds belong to this group being examined because of their cytotoxic, hepatoprotective, cardiovascular, antioxidant action of B and I cucurbitacins [21].

Song et al. [22] isolated two cucurbitacins from the ethyl acetate extract of *C. colocynthis* fruit, but these were not evaluated against insects. Similarly, Ding et al. [23] identified a mixture of spinasterol, 22,23-dihydrospinasterol from the roots of *Bermeuxia thibetica,* but its bioactivity was also not evaluated. However, Sinha et al. [24] reported that spinasterol, 22,23-dihydrospinasterol exhibited by *Melothria maderaspatana* showed biologic activities. It has been reported that Artemisia extracts contain valuable phytochemicals, possess insecticidal activity which were mainly attributed to the presence of fernenol and other phytoconstituents [25]. Furthermore, constituents of *Artemisia vulgaris* like psilostachy C, Maackiain, psilostachyin A and fernenol possess medicinal as well as antibacterial activities and also used by farmers for the preservation of crops and stored grains products [26].

Although some studies have been conducted on isolation and identification of various compounds including spinasterol, 22,23-dihydrospinasterol and fernenol from natural plant resources, their use as insecticidal purposes especially against *B. brassicae* has not been evaluated so far. Therefore, the present novel study was conducted in a comprehensive way for the isolation, characterization and evaluation of biochemical compounds from *C. colocynthis* leaves against *B. brassicae.*

## 2. Results

### 2.1. Preliminary Toxicity Evaluations

Interestingly, all solvents extract caused significant mortality of *B. brassicae*. However, maximum mortality (85%) was recorded at 96 h post exposure by methanol extract, followed by ethanol and chloroform extract (80%) and (76.67%), respectively. In contrast, at 72 h exposure, maximum mortality afforded by methanol extract was 66.67%, followed by chloroform and ethanol extract (58.33%) and (53.33%), respectively. However, minimum mortality was recorded via distilled water extract which was 35% and 18.33% at 96 and 72 h exposure, respectively. The results also revealed that mortality is dependent upon concentration and prolonged time exposure (Figure 1).

### 2.2. Extraction, Separation and Purification of Extract

Powdered leaves of *C. colocynthis* were extracted with different polarity based solvents. Fractionation of the extract was performed by chromatographic techniques. Collected fractions followed by separation by silica gel column were concentrated and weighed following their Thin Layer Chromatography (TLC) analysis (Table 1). Out of 8 of the obtained fractions, D_2_ fraction was further purified by repeated column chromatography on silica gel column, Sephadex gel column and preparative TLC and finally two compounds referred to as D2 (N) and D3A bearing weight 35 mg and 21 mg, respectively were purified.

### 2.3. Mass Spectrum and Elemental Analysis

The C, H and N elemental analysis is helpful for the calculation of empirical formula. The mass spectrometer examination recorded the molecular ion peak (M) at *m*/*z* 426, 414 and 412. These data suggested the expected molecular formula as C_30_H_50_O (Figure 2) and a mixture of C_29_H_50_O, C_29_H_48_O (Figure 3) for D3A and D2 (N), respectively. However, D2 (N) was a mixture of two compounds referred to as D2 (N1) and D2 (N2) and the ratio of these two compounds was recorded as 5:4 from the mass spectrum and nuclear magnetic resonance analysis.

### 2.4. NMR Analysis (^1^H-NMR and ^13^C-NMR)

The chemical structures of the purified compounds D2 (N) and D3A were characterized by nuclear magnetic resonance (NMR), Bruker BioSpin, Billerica MA for 1D (^1^H-NMR, ^13^C-NMR), mass spectrum and elemental analysis. All NMR experiments were carried out at room temperature.

In the ^1^H-NMR and ^13^C-NMR spectrum of the compound D_2_ (N 1) ^1^H-NMR (600 MHz, CDCl_3_) *δ* 5.16 (m, 1H, H-7), 3.60 (m, 1H, H-3) allylic protons, 0.93 (d, *J* = 7.0 Hz, 3H, -CH_3_), 0.85 (t, *J* = 6.8 Hz, 3H, -CH_3_), 0.82 (d, *J* = 6.9 Hz, 3H, -CH_3_), 0.81 (d, *J* = 6.9 Hz, 3H, -CH_3_), 0.80 (s, 3H, -CH_3_), 0.55 (s, 3H, -CH_3_) (Figure 4a). ^13^C-NMR (151 MHz, CDCl_3_) *δ* 11.8, 12.0, 13.0, 18.8, 19.0, 19.7, 21.5, 22.9, 23.0, 26.1, 27.9, 29.1, 29.6, 31.4, 33.8, 34.1, 36.5, 37.1,37.9, 39.5, 40.2, 43.3, 45.8, 49.4, 55.0, 56.0, 71.0, 117.3, 139.5, (Spinasterol, C_29_H_50_O; 414.0 g/mol) (Figure 4b). However, ^1^H-NMR and ^13^C-NMR spectra of the compound D_2_ (N 2) ^1^H-NMR (600 MHz, CDCl_3_) *δ* 5.16 (overlap, 1H H-7), 5.16 (overlap, 1H, H-22), 5.03 (dd, *J* = 15.1, 8.8 Hz, 1H, H-23), 3.60 (m, 1H, H-3), ) the allylic protons, 1.03 (d, *J* = 7.0 Hz, 3H, -CH_3_), 0.85 (t, *J* = 6.8 Hz, 3H, -CH_3_), 0.82 (d, *J* = 6.9 Hz, 3H, -CH_3_), 0.81 (d, *J* = 6.9 Hz, 3H, -CH_3_), 0.80 (s, 3H, -CH_3_), 0.54 (s, 3H, -CH_3_) (Figure 4). ^13^C-NMR (151 MHz, CDCl_3_) *δ* 11.9, 12.2, 13.0, 18.9, 21.0, 21.3, 21.5, 22.9, 25.3, 28.4, 29.6, 31.4, 31.8, 34.1, 37.1, 37.9, 39.4, 40.2, 40.8, 43.2, 49.4, 51.2, 55.0, 55.8, 71.0, 117.4, 129.4, 138.1, 139.5, (22,23-dihydrospinasterol, C_29_H_48_O; 412.0 g/mol) (Figure 4b).

In the ^1^H-NMR and ^13^C-NMR spectra of the compound D_3_A, ^1^H-NMR (600MHz, CDCL_3_) δ 5.30 (s, 1H, C=CH), 3.26–3.17 (m, 1H, -OH), the allylic protons, 1.07 (s, 3H, -CH_3_), 0.96 (s, 3H, -CH_3_), 0.89 (d, *J* = 6.4 Hz, 3H, -CH_3_), 0.87 (s, 3H, -CH_3_), 0.84–0.80 (m, 6H, two -CH_3_), 0.76 (s, 3H, -CH_3_), 0.73 (s, 3H, -CH_3_), 0.73–2.03 (m, 48H, including 8 CH_3_) (Figure 4c). ^13^C-NMR (151 MHz, CDCL_3_) δ: 13.90, 15.77, 17.87, 19.07, 20.05, 22.04, 22.98, 25.13, 27.37, 28.13, 28.06, 29.20, 29.63, 30.69, 36.07, 36.65, 37.57, 37.70, 39.19, 39.27, 39.90, 40.96, 42.85, 44.20, 50.29, 51.88, 59.58, 79.07, 116.11, 150.98, (Fernenol; C_30_H_50_O; 426.7 g/mol) (Figure 4d).

### 2.5. Structures Elucidation of the Compounds

Chemical structures of the identified fractions of D3A and D2 (N) were characterized by ^1^H-NMR and ^13^C-NMR spectroscopy and their molecular formulae were predicted by the mass spectrum. D3A compound was identified as fernenol (Figure 5). Another purified compound was a little bit impure and due to this impurity, the ^1^H-NMR spectrum was overlapped and hence, mixture of two compounds was identified as Spinasterol and 22, 23-dihydrospinasterol from the fraction D2 (N) with ratio 5:4 in the mixture (Figure 6a,b).

### 2.6. Bioassay Study

Toxicity of the identified compounds was evaluated extensively against *B. brassicae* by contact, residual and greenhouse assay and data were collected after a specific period of time exposure.

#### 2.6.1. In Vitro Contact Toxicity

The mortality data described in Table 2 exposed the contact efficacy of the spinasterol, 22,23-dihydrospinasterol and fernenol against *B. brassicae*. It was observed that the percent mortality rate of aphids was directly associated with the concentration and period of exposure. Results indicated that extreme mortality recorded was 66.67% and 56.66% at 72 and 48 h exposure, respectively by spinasterol, 22,23-dihydrospinasterol at 50 µg/mL. In contrast, moderate mortality was recorded by fernenol which was 30.00% and 21.67% at 72 and 48 h exposure, respectively, at the same concentration.

#### 2.6.2. In Vitro Residual Toxicity

The mortality data described in Table 3 exposed the residual efficacy of spinasterol, 22,23-dihydrospinasterol and fernenol against *B. brassicae*. Results demonstrated that highest mortality (53.33%) was observed at exposure of 72 h followed by (43.33%) at 48 h by spinasterol, 22,23-dihydrospinasterol at the concentration of 50 µg/mL. On the other hand, fernenol afforded lower mortality, i.e., (23.33%) and (16.67%) at the same exposure time and concentration, respectively.

#### 2.6.3. In Vivo Assay

The mortality data presented in Table 4 revealed the insecticidal activity of the purified compounds against *B. brassicae* under greenhouse conditions. Results showed that mortality rate of aphids under greenhouse conditions was 60.00% at 72 h exposure via spinasterol, 22,23-dihydrospinasterol at 50 µg/mL. In contrast, mortality recorded by fernenol was 25.00% at 50 µg/mL concentration and 72 h exposure.

Probit analysis exposed the LC_50_, slope value, Chi-square and fiducial limits at 95% confidence interval. Lowest LC_50_ values recorded by spinasterol, 22,23-dihydrospinasterol were 32.36 and 44.79 µg/mL after 72 h and 48 h of exposure period, respectively via contact assay. In contrast, at the same exposure period, LC_50_ values by residual assay were 44.57 and 58.38 µg/mL, respectively. On the other hand, LC_50_ values recorded by greenhouse assay were 37.50 and 48.90 µg/mL at 72 and 48 h exposure, respectively (Table 5). Similarly, LC_50_ values recorded via fernenol were 47.99 and 63.08 µg/mL; 57.46 and 104.46 µg/mL; 58.67 and 76.52 µg/mL at 72 and 48 h by contact, residual and greenhouse assay, respectively (Table 6).

## 3. Discussion

Because of the problems associated with the use of chemical pesticides for pest administration, the introduction of natural products particularly from plants source is pressing. Essential oils or extracts of plant origin are commonly used for plant protection measures because of their effectiveness against different life stages of insect pests. However, screening of suitable candidate plants for isolation, purification and identification of active ingredients is very crucial [27]. This technique was adopted to obtain active compounds from *C. colocynthis* leaves. Additionally, different techniques are used in the extraction, separation and purification of bioactive compounds from natural plant resources.

In our results for the preliminary toxicity evaluations of the solvents extract showed significant mortality of *B. brassicae.* Among the solvents extracts, methanol extract afforded high mortality which is in accordance with [28,29,30] who reported maximum mortality of *Aphis craccivora* by methanolic extract of *C. colocynthis* followed by ethyl acetate and petroleum ether extract. However, the activity of the crude extract can be attributed to the existence of specific chemical compounds of the plants like fatty acids (linoleic and oleic acid) glycosides (flavonoids, phenols, saponins), terpenoids and alkaloids, etc. [31]. Recently, Ahmed et al. [32] reported that solvents (methanol, ethanol, ethyl acetate, chloroform, acetone and hexane) extract of *C. colocynthis* leaves exhibited important phytochemicals such as alkaloids, glycosides, steroids, saponins, phenol, tannins and flavonoids along with potential antioxidant activities however, acetone and ethanol extract displayed as potent antioxidants. Further, these solvents extract also exhibited pronounced insecticidal activities [33]. Butanol fraction of the methanol extract from the *C. colocynthis* plant possessed insecticidal properties because of the presence of flavone glucosides and two cucurbitacins glucosides [19] Moreover, Yoshikawa et al. [34] described that alcoholic extract obtained from *C. colocynthis* fruits contains various compounds comprising cucurbitacins E 2-*O*-a-d-glucopyranoside, colocynthosides A and B, aglycon and cucurbitacins E. Among the isolated two triterpens glycosides and four cucurbitacins from *C. colocynthis* leaves, one was found effective against colon cancer cells HT29 and Caco-2 [35]. Plants possess a wide range of biologic compounds involved in their mechanism of chemical defense. These natural products contain significant prospective against variety of insects [36,37]. Activity of some of the natural compounds was evaluated against *Blattellea germanica* which showed LC_50_ values as 0.07 mg cm^−1^, 0.06 mg cm^−1^ and 0.07 mg cm^−1^ via camphor, pulegone and verbenone, respectively [38]. Similarly, eugenol, isoeugenol, carvecol, thymol and p-cymene had shown anti-adulticidal activity at 1 mg adult^−1^ against *B. germanica* [39].

As different techniques are employed in the extraction, separation, purification and identification of bioactive compounds thus, following these techniques, two compounds were identified as; a mixture of spinasterol and 22,23-dihydrospinasterol and other pure compound fernenol (d:c-friedo-b’:a’-neogammacer-9(11)-en-3alpha-olfern-9(11)-en-3alpha-o). Spinasterol is contained by Cucurbitaceae, Stegnospermaceae, Phytolaccaceae and Polygalaceae. In a study Meneses-Sagrero et al. [40] identified spinasterol form the methanol extract of *Stegnosperma halimifolium* (B.) and evaluated against cancer cell line. It was reported that spinasterol exhibited potential activity as antiproliferative against two cell lines of cervical cancer such as HeLa and RAW 264.7. Studies have also demonstrated that spinasterol exhibited different biologic activities including antidepressant [41], anti-ulcerogenic [42], anti-inflammatory [43], and antiproliferative activities [44]. Spinasterol, 22,23-dihydrospinasterol possess pharmacological and cytotoxic exertions likewise, it was isolated from *Bougainvillea spectabillis* and exhibited strong inhibition of xanthine oxidase being IC_50_ of 39.21 µM [45].

Fernenol belongs to the class of organic compounds known as triterpenoids containing six isoprene units. Xian-xue W et al. [46] isolated this compound from the whole plant extract of *Arenaria polytrichoides*, however, its activity was not evaluated against pests. Similarly, Li et al. [47] isolated three terpenoids including fernenol from *Ainsliaea yunnanensis* showed cytotoxic effects on human acute monocytic Leukemia cell line with IC_50_ being 1.73 µM. Some studies revealed that fernenol play an important role in the inhibition of mycelial development of *Colletotrichum gloeosporioides* being 47.5 mg mL^−1^ EC_50_, and also significantly effective against anthracnose of mango when used at 100 mg L^−1^ and 200 mg L^−1^ concentration [48].

Our results demonstrated that spinasterol, 22,23-dihydrospinasterol exhibited aphicidal activity which caused significant mortality of this pest via different bioassays whereas, fernenol exhibited moderate aphicidal activity. Similar findings on extraction, purification and activity of purified compound 2-*O*-β-d-glucapyranosylcucurbitacin E was evaluated by Torkey et al. [49] against *Aphis craccivora* showed significant mortality with LC_50_ being 11,003 ppm. Moreover, aphicidal activity of *Eupatorium adenophorum* isolated compound 9-oxo-10, 11-dehydroageraphorone was evaluated against *Pseudoregma bambucicola* displayed 73.33% mortality at 2 mg mL^−1^ with 6 h exposure. Similarly, Nong et al. [50] recorded complete control of this pest at the same concentration after 30 days in a field experiment.

Thus, this unique and innovative research was performed first time for isolation, purification and characterization of bioactive compounds from *C. colocynthis* leaves and to investigate their insecticidal potential against cabbage aphid *B. brassicae*, which is an injurious pest of cabbage and other crops. It is worth mention that these compounds were isolated and evaluated from *C. colocynthis* for the first time against this pest.

## 4. Material and Methods

### 4.1. Collection of Plants and Aphids

*C. colocynthis* (Colocynth), locally recognized as Tumba, was the study plant. Leaves were collected from natural habitat of desert one climate of District Bahawalnagar, Pakistan with latitudinal and longitudinal gradients (29°59′34′’N, 73°15′13′’E) from March to April 2018. This area contains dry climate with an average precipitation of 204 mm annually and temperature ranges from 12.7–45 °C. The samples were authenticated at Entomological Research Institute Faisalabad, Pakistan.

Cabbage aphids were collected from wild cabbage plants from the surrounding field area of Shenyang Agricultural University, Shenyang China. During the collection of aphids it was insured that no pesticides was applied on the plants. The population of the aphids was maintained on cabbage plants grown in the greenhouse at 20 ± 5 °C and 45% ± 5% relative humidity (RH), along with a photoperiod of 16:8 (light: dark).

### 4.2. Extraction of Plant Material

Extraction was performed thrice at room temperature for three days by using the cold extraction/solvent extraction method. Different organic solvents were used for extraction purposes bearing varying polarity. Then, filter the contents and volume was reduced by concentrating the filtered material through the rotary evaporator (Model R-210, Buchi Switzerland, Flawil, Switzerland). Next, the filtrate was allowed to dry for 12 h in a fume hood at 28 °C and then, the dry extract was preserved in glass bottles at 4 °C for further experimentation.

### 4.3. Sample Preparations and Preliminary Toxicity Evaluations

Each of the extract obtained by using different solvents was evaluated against *B. brassicae* at 50 µg/mL concentration at an exposure of 6, 12, 24, 48, 72 and 96 h. For toxicity tests, dried extracts were dissolved in acetone (1 mL for each solvent extract) and mixed with 1% Tween-20 (prepared in distilled water) for preparing concentration. For control treatment, check (CK) was also prepared in 1% Tween-20 including acetone but excluding extract to prepare a 50 µg/mL concentration. Thus, to obtain homogenous mixture for separation and purification purposes, a 2-g sample from each solvent extract was taken in porcelain dish (14 g in total), dissolved in appropriate amount of dichloromethane and then added equal quantity of silica gel into it. The resulting mixture was left for overnight in a fume hood at 28 °C to evaporate the solvent and on drying; the residue was ground to fine powder by mortar and pestle and stored in glass stopper vials at 4 °C for further use.

### 4.4. Separation and Purification of Extract

The prepared sample was separated and purified by column chromatographic techniques. The dried sample was chromatographed on silica gel column (200–300 mesh). Gradient eluent of 300 mL each of the eluted fraction was collected using a gradient ratio of petroleum ether and ethyl acetate. The mobile phases used for silica gel column were PE:EA of 100:0, 100:2, 100:4, 100:8, 100:16, 100:32, 100:64, 100:100 and at the end methanol was used and finally, total 49 fractions were collected. Lastly, eight sub-fractions (A-H) were obtained followed by the mixing of the same polarity fractions. The mobile phases used for gel column Sephadex, LH-20 (40–120 µm) were methanol and dichloromethane with 1:1 ratio. For further purification preparative TLC plates were used and target compounds were scrapped, dissolved in ethyl acetate and concentrated on a rotary evaporator (Buchi Switzerland R-210, Flawil, Switzerland) for obtaining pure compounds with weight calculation.

### 4.5. Mass Spectrum and Elemental Analysis

Mass spectrum data were calculated on the Triple Quad GC-MS 7000C and Triple Quad LC/MS 6440 mass Spectrometer (Agilent Technologies, Santa Clara, CA, USA). Vario EL III element analyzer was used for elemental analysis (Elementar Analyssensysteme GmbH, Frankfurt, Germany).

### 4.6. Nuclear Magnetic Resonance (NMR) ^1^H-NMR and ^13^C-NMR Spectrum

To determine ^1^H-NMR spectra, Deuterated chloroform (CDCL_3_) was used as a solvent while for an internal standard, tetramethylsilane (TMS) was used. ^1^H-NMR spectra were evaluated on 300 MHz and 600 MHz spectrometer whereas, ^13^C-NMR spectra were assessed on a 151-MHz spectrometer (Bruker, Karlsruhe, Germany). Chemical shift values were recorded in parts per million (δ, ppm). The coupling constants value (J) was described in Hz. The splitting patterns of proton signals were also designated as follows: singlet (s), doublet (d), a doublet of doublets (dd), a doublet of the doublet of doublets (ddd), triplet (t), the quartet (q) and the multiplet (m)

### 4.7. Bioassay Study

Activity of the compound against *B. brassicae* was evaluated In vitro and In vivo. For In vitro assay, contact and residual toxicity methods were adopted while; In vivo experiment was conducted under greenhouse condition of Shenyang Agricultural University. Serial concentrations such as 6.25, 12.50, 25 and 50 µg/mL were prepared in Tween-20 (1%) solutions and replicated thrice.

#### 4.7.1. In Vitro Contact Toxicity

For contact assay, 10 adult wingless aphids were dipped in respective concentrations for 5 s and released on freshly cut cabbage leaves placed in glass petri dishes. Check (CK) was prepared in 1% Tween-20 solution, but without adding a purified compound. All of the prepared petri dishes including control were placed in an incubator at 20 ± 5 °C, 65% RH for three days along with 16:8 (light: dark) photoperiod.

#### 4.7.2. In Vitro Residual Toxicity

Fresh cabbage leaves were cut off and dipped for 10 s in respective concentration and dried in air for half an hour. Next, 10 adult wingless aphids were released on these leaves contained in glass petri dishes. Check (CK) was prepared in 1% Tween-20 solution, but without adding purified compound. All of the prepared petri dishes including check were placed in an incubator at 20 ± 5 °C, 65% RH for three days along with 16:8 (light: dark) photoperiod.

#### 4.7.3. In Vivo Toxicity

Cabbage plants were grown and maintained in the greenhouse for the rearing of aphids and bioassay study. Prior to experiment, plants were sprayed with water to remove impurities and left for half an hour to dry in the open environment. Then, 10 wingless adult aphids were released on cabbage plants at 6–8 true leaf stage. After one hour of releasing aphids, plants were sprayed with respective concentrations (1–2 showers) with hand pump/sprayer. A control treatment was sprayed with distilled water solution in 1% Tween-20. Treated plants along with control were left for three days in greenhouse conditions.

### 4.8. Data Collection and Statistical Analysis

Data on mortality at a specific time period of 12, 24, 48 and 72 h were recorded regularly by examination under the microscope. Those individuals who presented no response on probing by needle were recorded as dead.

Analysis of the mortality data were performed by using analysis of variance (ANOVA). The mean mortality difference among treatments was intended at *p* = 0.05 by Duncan multiple range test DMRT with IBM-SPSS statistics 25.0 version software. Probit analysis was performed using EPA Probit analysis program version 1.5.

## 5. Conclusions

The present investigations indicated that *C. colocynthis* possess potential botanical agents. Results also demonstrated that *B. brassicae* showed reasonable sensitivity to isolated spinasterol, 22,23-dihydrospinasterol via all bioassay. In contrast, fernenol displayed moderate toxicity against this pest. Additionally, the contact bioassay produced higher mortality than the residual bioassay. Therefore, these chemical compounds may be introduced as alternatives to synthetic chemical insecticides. However, additional research is necessary for the purification and characterization of more bioactive constituents and their appraisals against *B. brassicae* and other field crops insect pests.

## Figures and Tables

**Figure 1 molecules-25-02184-f001:**
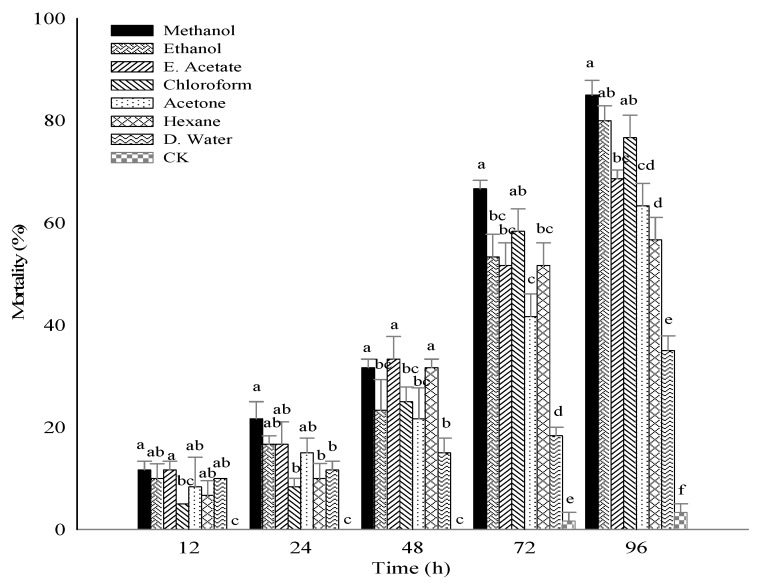
Mortality of *Brevicoryne brassicae* by *Citrullus colocynthis* leaves extracts. Values are represented as mean ± standard deviation followed by different superscripts (a, b, c, d, e, f, ab, bc, cd) were not significantly different according to Duncan multiple range test (DMRT) at *p* > 0.05. E. Acetate (ethyl acetate); D. Water (distilled water); CK (check in distilled water); Time (h).

**Figure 2 molecules-25-02184-f002:**
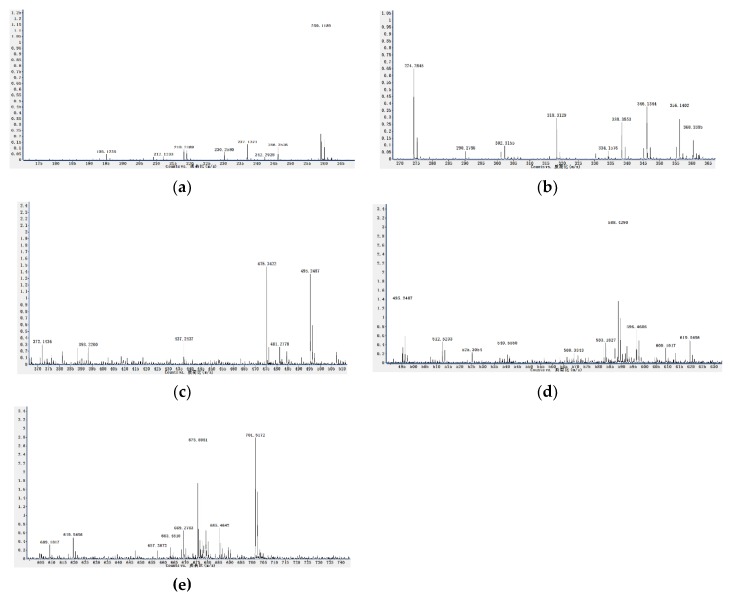
(**a**–**e**) The full mass spectrum for the purified compound, D3A from *Citrullus colocynthis* leaves.

**Figure 3 molecules-25-02184-f003:**
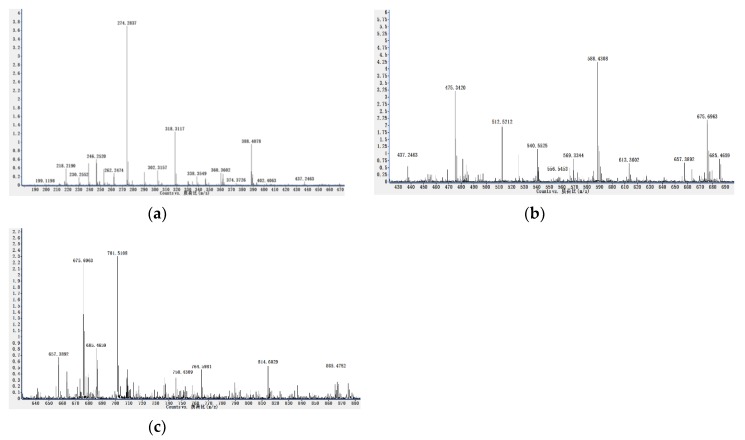
(**a**–**c**) The full mass spectrum for the purified compound, D2 (N) from *Citrullus colocynthis* leaves.

**Figure 4 molecules-25-02184-f004:**
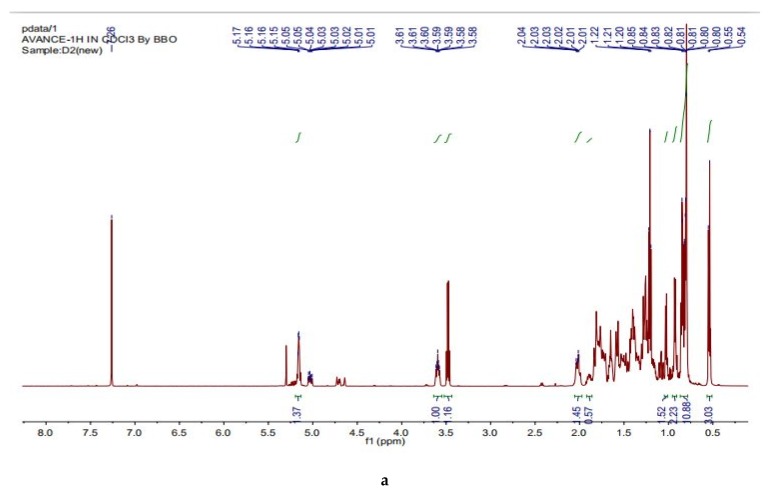
(**a**). ^1^H-NMR spectrum of the compound D2 (N); (**b**). ^13^C-NMR spectrum of compound D2 (N); (**c**). ^1^H-NMR spectrum of the compound D3A; (**d**). ^13^C-NMR spectrum of the compound D3A.

**Figure 5 molecules-25-02184-f005:**
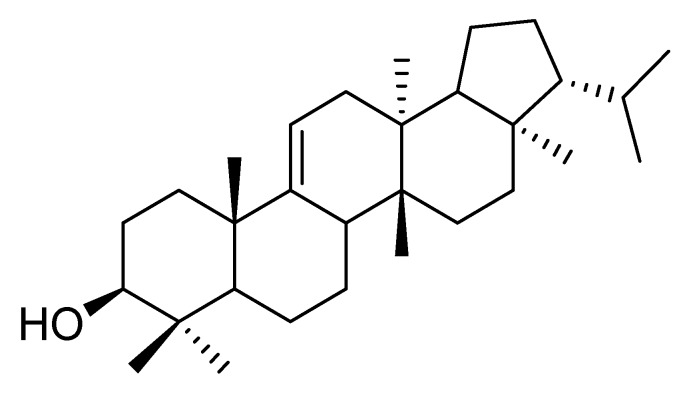
Chemical structure of compound; Fernenol from D3A.

**Figure 6 molecules-25-02184-f006:**
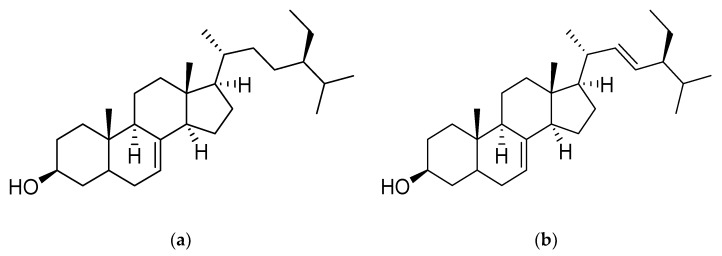
Chemical structures of compound; (**a**). Spinasterol; (**b**) 22, 23-dihydrospinasterol from D2 (N).

**Table 1 molecules-25-02184-t001:** Obtained fractions by silica gel column.

Eluted Fractions	Collected Fractions	Weight (g)
1–10	A	0.293
11–13	B	0.355
14–23	C	0.796
24–27	D	0.662
28–30	E	0.710
31–34	F	0.270
35–40	G	0.447
41–43	H	0.428
44–49	I	2.841

**Table 2 molecules-25-02184-t002:** Mortality of *Brevicoryne brassicae* by spinasterol, 22,23-dihydrospinasterol and fernenol via contact assay.

Conc. (µg/mL)	Mean Mortality (%) with Time (h)							
	12		24		48		72	
	Spinasterol, 22,23-Dihydr- Spinasterol	Fernenol	Spinasterol, 22,23-Dihydr- Spinasterol	Fernenol	Spinasterol, 22,23-Dihydr- Spinasterol	Fernenol	Spinasterol, 22,23-Dihydr- Spinasterol	Fernenol
6.25	0.00 ± 0.00^c^	0.00 ± 0.00^a^	0.00 ± 0.00^c^	0.00 ± 0.00^c^	3.33 ± 3.33^c^	1.67 ± 1.67^c^	6.67 ± 3.33^c^	3.33 ± 1.67^c^
12.50	3.33 ± 3.33^c^	0.00 ± 0.00^a^	10.00 ± 0.00^c^	5.00 ± 2.87^bc^	13.3 ± 3.33^c^	6.67 ± 1.67^bc^	20.00 ± 5.77^c^	10.0 ± 2.89^bc^
25	16.67 ± 3.33^b^	3.33 ± 1.67^a^	23.33 ± 3.33^b^	8.33 ± 1.67^ab^	26.67 ± 3.33^b^	11.67 ± 1.67^b^	43.33 ± 6.67^b^	15.0 ± 2.89^b^
50	26.67 ± 3.33^a^	3.3 ± 1.67^a^	40.00 ± 0.00^a^	13.3 ± 1.67^a^	56.67 ± 3.33^a^	21.67 ± 1.67^a^	66.67 ± 3.33^a^	30.002.89^a^
CK	0.00 ± 0.00^c^	0.00 ± 0.00^a^	0.00 ± 0.00^c^	0.00 ± 0.00^c^	3.33 ± 3.33^c^	1.67 ± 1.67^c^	3.33 ± 3.33^ed^	3.33 ± 1.67^c^
Statics Summary								
S.S	1693	40.00	3506	390.0	5960	840.0	8573	1460
df	4	4	4	4	4	4	4	4
M.S	423	10.00	876.6	97.50	1490	210.0	2143	365.0
F	21.17***	3.00^NS^	131.50***	11.70***	44.70***	25.20***	32.150***	19.91***

Values are described as mean ± standard error followed by different superscripts were significantly different according to (Duncan multiple range test DMRT *p* > 0.05). S.S (Sum of square); df (Degree of freedom); M.S (Mean square); F (Significance); CK (Check); NS (Non-significant); *** (Highly significant).

**Table 3 molecules-25-02184-t003:** Mortality of *Brevicoryne brassicae* by spinasterol, 22,23-dihydrospinasterol and fernenol via residual assay.

Conc. (µg/mL)	Mean Mortality (%) with Time (h)							
	12		24		48		72	
	Spinasterol, 22,23-Dihydro- Spinasterol	Fernenol	Spinasterol, 22,23-Dihydro- Spinasterol	Fernenol	Spinasterol, 22,23-Dihydro- Spinasterol	Fernenol	Spinasterol, 22,23-Dihydro- Spinasterol	Fernenol
6.25	0.00 ± 0.00^b^	0.00 ± 0.00^a^	0.00 ± 0.00^c^	0.00 ± 0.00^b^	3.33 ± 3.33^c^	0.00 ± 0.00^c^	6.67 ± 3.33^c^	1.67 ± 1.67^c^
12.50	0.00 ± 0.00^b^	0.00 ± 0.00^a^	3.33 ± 3.33^c^	1.67 ± 1.67^b^	6.67 ± 3.33^c^	3.33 ± 1.67^bc^	13.33 ± 3.33^c^	5.00 ± 0.00^bc^
25	6.67 ± 3.33^b^	1.67 ± 1.67^a^	13.33 ± 3.33^b^	5.00 ± 2.89^b^	26.67 ± 3.33^b^	6.67 ± 1.67^b^	36.676 ± 3.33^b^	10.00 ± 2.89^b^
50	13.33 ± 3.33^a^	1.67 ± 1.67^a^	23.33 ± 3.33^a^	11.67 ± 1.67^a^	43.33 ± 3.33^a^	16.67 ± 1.67^a^	53.33 ± 3.33^a^	23.33 ± 1.67^a^
CK	0.00 ± 0.00^b^	0.00 ± 0.00^a^	0.00 ± 0.00^c^	0.00 ± 0.00^b^	0.00 ± 0.00^c^	3.33 ± 1.67^bc^	3.33 ± 3.33^c^	3.33 ± 3.33^c^
Statics summary								
S.S	426.6	10.00	1240	290.0	4093	493.3	5560.0	923.3
df	4	4	4	4	4	4	4	4
M.S	106.67	2.50	310.0	72.50	1023	123.3	1390	230.8
F	8.00**	0.750*	15.00***	8.70***	38.37***	18.50***	41.70***	23.08***

Values are described as mean ± standard error followed by different superscripts were significantly different according to (Duncan multiple range test DMRT *p* > 0.05). S.S (Sum of square); df (Degree of freedom); M.S (Mean square); F (Significance); CK (Check); *** (Highly significant); ** (Highly significant); * (Significant).

**Table 4 molecules-25-02184-t004:** Mortality of *Brevicoryne brassicae* by spinasterol, 22,23-dihydrospinasterol and fernenol via greenhouse assay.

Conc. (µg/mL)	Mean Mortality (%) with Time (h)							
	12		24		48		72	
	Spinasterol, 22,23-Dihydro- Spinasterol	Fernenol	Spinasterol, 22,23-Dihydro- Spinasterol	Fernenol	Spinasterol, 22,23-Dihydro- Spinasterol	Fernenol	Spinasterol, 22,23-Dihydro- Spinasterol	Fernenol
6.25	0.00 ± 0.00^b^	0.00 ± 0.00^a^	0.00 ± 0.00^b^	0.00 ± 0.00^c^	0.00 ± 0.00^c^	0.00 ± 0.00^d^	3.33 ± 3.33^cd^	5.00 ± 2.89^c^
12.50	3.33 ± 3.33^b^	0.00 ± 0.00^a^	6.67 ± 3.33^b^	3.33 ± 1.67^bc^	10.00 ± 0.00^c^	5.00 ± 0.00^c^	13.33 ± 3.33^c^	6.67 ± 1.67^c^
25	10.00 ± 0.00^b^	3.33 ± 1.67^a^	20.00 ± 5.77^a^	6.67 ± 1.67^ab^	23.33 ± 3.33^b^	10.00 ± 0.00^b^	40.00 ± 0.00^b^	15.00 ± 2.89^b^
50	20.00 ± 5.77^a^	3.33 ± 1.67^a^	30.00 ± 5.77^a^	10.00 ± 2.89^a^	50.00 ± 5.77^a^	18.33 ± 1.67^a^	60.00 ± 5.77^a^	25.00 ± 2.89^a^
CK	0.00 ± 0.00^b^	0.00 ± 0.00^a^	0.00 ± 0.00^b^	0.00 ± 0.00^c^	3.33 ± 3.33^c^	1.67 ± 1.67^d^	3.33 ± 3.33^c^	3.33 ± 1.67^c^
Statics summary								
S.S	866.6	40.00	2106.6	226.6	4960.0	656.6	8000	976.6
df	4	4	4	4	4	4	4	4
M.S	216.6	10.00	526.67	56.67	1240.0	164.1	2000.0	244.1
F	8.13**	3.00*	11.29**	6.80***	37.20***	49.25***	60.00***	13.32***

Values are described as mean ± standard error followed by different superscripts were significantly different according to (Duncan multiple range test DMRT *p* > 0.05). S.S (Sum of square); df (Degree of freedom); M.S (Mean square); F (Significance); CK (Check); *** (Highly significant); ** (Highly significant); * (Significant).

**Table 5 molecules-25-02184-t005:** Probit analysis of the effects of spinasterol, 22,23-dihydrospinasterol on *Brevicoryne brassicae.*

Bioassay	Time (h)	LC_50_ (µg/mL)	95% F.L		Slope ± SE	χ^(2)^
			Lower	Upper		
Contact	12	87.07	52.59	437.6	2.039 ± 0.60	1.75
	24	61.04	41.42	151.1	2.089 ± 0.52	1.06
	48	44.79	32.68	82.15	2.478 ± 0.72	0.31
	72	32.36	23.66	48.39	2.322 ± 0.54	0.12
Residual	12	143.5	69.10	1634	1.272 ± 1.01	0.77
	24	104.1	57.74	1003	1.927 ± 0.62	1.24
	48	58.38	39.46	143.7	1.981 ± 0.49	0.49
	72	44.57	30.99	90.95	2.078 ± 0.59	0.42
G. house	12	133.4	65.40	6044	1.870 ± 0.68	0.33
	24	83.23	50.25	387.1	1.920 ± 0.55	1.09
	48	48.90	36.29	86.10	2.462 ± 0.54	0.78
	72	37.50	27.86	57.92	2.502 ± 0.63	0.71

F.L; Fiducial limits. χ^(2)^; Chi-square.

**Table 6 molecules-25-02184-t006:** Probit analysis of the effects of fernenol on *Brevicoryne brassicae*.

Bioassay	Time (h)	LC_50_ (µg/mL)	95% F.L		Slope ± SE	χ^(2)^
			Lower	Upper		
Contact	12	345.2	89.21	1145	1.89 ± 0.73	1.62
	24	135.0	61.44	6531	0.91 ± 1.42	1.65
	48	63.08	40.63	296.3	1.47 ± 1.02	0.39
	72	47.99	32.59	94.62	1.27 ± 1.11	0.66
Residual	12	775.9	79.10	1634	1.43 ± 1.19	0.82
	24	109.7	60.35	1456	2.05 ± 0.69	0.24
	48	104.4	57.08	1031	1.82 ± 0.58	0.71
	72	57.46	44.78	221.1	3.12 ± 1.19	0.24
G. house	12	345.2	98.43	1316	1.61 ± 0.94	0.61
	24	158.0	67.56	2353	1.52 ± 0.57	0.99
	48	76.72	49.39	1781	2.33 ± 0.95	2.39
	72	58.67	38.68	282.9	2.13 ± 0.89	0.22

F.L; Fiducial limits. χ^(2)^; Chi-square.

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
