# Peer review of "Spinasterol, 22,23-Dihydrospinasterol and Fernenol from Citrullus Colocynthis L. with Aphicidal Activity against Cabbage Aphid Brevicoryne Brassicae L."

_molecules, 2020, doi:10.3390/molecules25092184_

Round 1

Reviewer 1 Report

The work presented in this manuscript found toxicity of some compounds extracted and purified from Citrullus colocynthis leaves to B. brassicae. The insecticidal activity of compounds found in this plant has extensively examined and reported to date, as mentioned by the present authors in line 60-64, and the present work adds a finding on aphicidal activity. In this respect, this work may be useful but not innovative.

General comment

  1. The authors have a priori, simplistic belief that compounds derived from natural products are safe and do not have toxicity to other animals including humans, which cannot be accepted. Such descriptions found throughout the manuscript should be deleted before it is published.
  2. Chemistry part of this work is well done.
  3. Toxicity test part of this work needs evaluation and discussion about the results. Do the obtained LC50 values superior to others?

Specific comments

  1. Percentage, LC50 and other numbers in Tables and text are expressed with 4- or 5-digit throughout the manuscript. Too much digit is meaningless and quite strange.
  2. Description in 2.1 (Line 74---) and Fig. 1 are not readily understood. Explanations are short. For instance, the authors should not mention “methanol” but “methanol extract of leaves” in this section. Some description should be added to help understand what is mentioned in this part of Result.
  3. 1 does not have title. No explanation of abbreviations D water and CK in the legend.
  4. 1, In what solvent were the extracts of different solvents, which were dried after extraction, dissolved and given to aphids?

Author Response

Response to Reviewer 1

Comments and Suggestions for Authors

The work presented in this manuscript found toxicity of some compounds extracted and purified from Citrullus colocynthis leaves to B. brassicae. The insecticidal activity of compounds found in this plant has extensively examined and reported to date, as mentioned by the present authors in line 60-64, and the present work adds a finding on aphicidal activity. In this respect, this work may be useful but not innovative.

Specific comments

Point 1: Percentage, LC50 and other numbers in Tables and text are expressed with 4- or 5-digit throughout the manuscript. Too much digit is meaningless and quite strange.

Response 1: LC50, %age values and other numbers in tables, figures and text are reduced to 2-4 digits.

Point 2: Description in 2.1 (Line 74---) and Fig. 1 is not readily understood. Explanations are short. For instance, the authors should not mention “methanol” but “methanol extract of leaves” in this section. Some description should be added to help understand what is mentioned in this part of result.

Response 2: This part was explained more and mortality percentage afforded by different solvents extract was described. Description of aphid mortality by various solvent extract was explained in detail.

Point 3: 1 does not have title. No explanation of abbreviations D water and CK in the legend.

Response 3: A title was given to Figure 1. Mortality of Brevicoryne brassicae by solvent extracts of C. colocynthis leaves. Similarly, D water and CK in the legend are explained as E. Acetate (Ethyl Acetate); D. Water (Distilled Water); CK (Check).

Point 4: In what solvent were the extracts of different solvents, which were dried after extraction, dissolved and given to aphids?

Response 4: For toxicity tests, dried extract was dissolved in their own solvents (1 mL) by which they were extracted and to prepare concentration, 1% tween-20 prepared in distilled water was used.

Reviewer 2 Report

Line 18.  The following sentences should be removed from abstract (Petroleum ether and ethyl acetate was used for silica gel column while, dichloromethane and methanol (1:1) was used for gel column (Sephadex LH-20). Compounds were further purified by using preparative TLC).

Line 33: spinasterol, 22,23-dihydrospinasterol is  (should be are )

Line 76: Interestingly,...... 85% was recorded at 96 h (post) exposure by methanol followed by ethanol and chloroform (extract)80,
and 76.67% respectively. Whereas, at 72 h exposure, maximum mortality afforded by methanol (extract).

line 94: Elemental analysis is beneficial for pure and very dry compounds, but none of the desribed compounds is pure as indicated in 1H-NMR.

line 109 : C-NMR (600 MHz, CDCl3)  should be (151 MHz)

line 114: C-NMR (600 MHz, CDCl3) should be (151 MHz)

Line 118: the H-NMR data are not as depicted in in the spectra....3.2 (1H, H-3 (HOCH) ) and the allylic proton are missing. (It has to be checked)

line 205-207 (not informative at all): Extraction may be able to be performed as solvent extraction, microwave assisted extraction and soxhelt extraction etc. However, for separation and purification, silica gel column chromatography, gel chromatography, preparative TLC and liquid phase preparations are commonly used.

Line 214: rephrase the sentence ; Our findings were further supported by Ahmed..........

line 216: Delazar, A.; Gibbons, S.; Kosari, A.R.; Nazemiyeh, H.; Modarresi, M.; Nahar, L.; Sarker, ......alot of names (Delazar et al is enough)

Line : rephrase the sentence; Moreover, Nong, et al. [41] 249
reported the aphicidal activity
.………...

line 288: use column chromatography instead of Glass chromatography..

line 289: gradient eluent instead stepwise elution...

line 305:13C-NMR spectra were assessed on 600MHz and 150 MHz, remove 600 MHz

Check reference 37.

Author Response

Response to Reviewer 2

Point 1: Line 18.  The following sentences should be removed from abstract (Petroleum ether and ethyl acetate was used for silica gel column while, dichloromethane and methanol (1:1) used for gel column (Sephadex LH-20). Compounds were purified by using preparative TLC).

Response 1: Said sentence was removed from the abstract.

Point 2: Line 33: spinasterol, 22,23-dihydrospinasterol is  (should be are)

Response 2: changed as “spinasterol, 22,23-dihydrospinasterol are more effective against

Point 3: Line 76: Interestingly,...... 85% was recorded at 96 h (post) exposure by methanol followed by ethanol and chloroform (extract)80, and 76.67% respectively. Whereas, at 72 h exposure, maximum mortality afforded by methanol (extract).

Response 3: has been followed and described as Interestingly, all solvents extract caused significant mortality of B. brassicae however; maximum mortality (85%) was recorded at 96 h post exposure by methanol extract followed by ethanol and chloroform extract (80%) and (76.67%) respectively. Whereas, at 72 h exposure, maximum mortality afforded by methanol extract was (66.67%) followed by chloroform and ethanol extract (58.33%) and (53.33%) respectively. However, minimum mortality was recorded via distilled water extract which was (35%) and (18.33%) at 96 and 72 h exposure respectively (Figure 1) The results also revealed that mortality is dependent upon concentration and prolonged time exposure.

Point 4: line 94: Elemental analysis is beneficial for pure and very dry compounds, but none of the described compounds is pure as indicated in 1H-NMR.

Response 4: Elemental analysis proposed two pure compounds. however, fernenol was a pure compound with molecular formula C30H50O  whereas, spinasterol, 22,23-dihydrospinasterol was a mixture of two compounds with molecular formula as  C29H50O, C29H48O.

Point 5: line 109: C-NMR (600 MHz, CDCl3) should be (151 MHz)

Response 5: Changed to 151 MHz Point 6: line 114: C-NMR (600 MHz, CDCl3) should be (151 MHz)

Response 6: Changed to 151 MHz Point 7: Line 118: the H-NMR data are not as depicted in in the spectra....3.2 (1H, H-3 (HOCH) ) and the allylic proton are missing. (It has to be checked).

Response 7: A few changes are done as, 1H-NMR (600MHz, CDCL3) δ 5.30 (s, 1H, C=CH), 3.26–3.17 (m, 1H, -OH), 1.07 (s, 3H, -CH3), 0.96 (s, 3H, -CH3), 0.89 (d, J= 6.4 Hz, 3H, -CH3), 0.87 (s, 3H, -CH3), 0.84–0.80 (m, 6H, two -CH3), 0.76 (s, 3H, -CH3), 0.73 (s, 3H, -CH3), 0.73–2.03 (m, 48H, including 8 CH3).

Point 8: line 205-207 (not informative at all): Extraction may be able to be performed as solvent extraction, microwave assisted extraction and soxhelt extraction etc. However, for separation and purification, silica gel column chromatography, gel chromatography, preparative TLC and liquid phase preparations are commonly used.

Response 8: Said statement was deleted.

Point 9: Line 214: rephrase the sentence; Our findings were further supported by Ahmed..........

Response 9: Sentence was rephrased as; Recently, Ahmed, et al. [27] reported that solvent extract of C. colocynthis leaves exhibited important phytochemicals further, it possess potential antioxidant and insecticidal activities [28].

Point 10: line 216: Delazar, A.; Gibbons, S.; Kosari, A.R.; Nazemiyeh, H.; Modarresi, M.; Nahar, L.; Sarker, ......alot of names (Delazar et al is enough)

Response 10: Described as Delazar et al. [19].

Point 11: Line : rephrase the sentence; Moreover, Nong, et al. [41] 249 reported the aphicidal activity …

Response 11: sentence was rephrased as; Moreover, aphicidal activity of Eupatorium adenophorum isolated compound 9-oxo-10, 11-dehydroageraphorone was evaluated against Pseudoregma bambucicola showed 73.33% mortality at 2mgmL-1 with 6 h exposure. Similarly, at the same concentration complete control of this pest was recorded after 30 days in field experiment.

Point 12: line 288: use column chromatography instead of Glass chromatography..

Response 12: Instructions followed and glass chromatography changed to column chromatography

Point 13: line 289: gradient eluent instead stepwise elution...

Response 13: stepwise elution changed to gradient eluent

Point 14:line 305:13C-NMR spectra were assessed on 600MHz and 150 MHz, remove 600 MHz

Response 14: 600 MHz removed from the said line

Point 15: Check reference 37.

Response 15: Corrected according to journal requirement

Reviewer 3 Report

The objectives of the presented study was to aimed to purify and identify the active compounds from Citrullus colocynthis leaves in order to test their efficacy against B. brassicae. The study seems to provide some interesting data on the topic. The study can be interesting for the Journal after revisions. Below I provide some indicative examples for the revision.

Comments:

Abstract

L20: TLC: give also in full

L18: via – it is latin. Should be in italic. Check throughout the text.

L22: in vitro and in vivo – they are latin. Should be in italic. Check throughout the text.

L23: viz. ??

Introduction

Provide more convincing information about that the compounds that you purified were not previously tested against B. brassicae.

L38: ‘cabbage aphid’ and not ‘Cabbage aphid’. Check throughout the text.

L58 and L261: Beginning of the sentences. Do not start with abbreviation. Citrullus …

Results

L75: Delete ‘Toxicity results at 50 µg/mL concentration at an exposure of 6, 12, 24, 48, 72 and 96 h are presented in (Figure 1). Interestingly,’.

L78: … respectively (Figure 1).

Figure 1: Give full explanation for E. Acetate; D. Water and CK.

Figure 1: Give a title for this figure. Presented title is just a footnote.

Figures 2 and 3: The titles are not well informative. Give more information.

Figure 4-7 Should be combined as Figure 4 A, B, C, D.

Tables 2-9. Give full name of B. brassicae. df instead of D.F. Give also in full (in the footnote S.S. df, MS, and F).

Tables 2 and 3 are the same topic. Should combined into one table.

Tables 4 and 5 are the same topic. Should combined into one table.

Tables 6 and 7 are the same topic. Should combined into one table.

L197: Discussion

Need more comparisons between the present results of this study and the previous studies on the compound efficacy of existing chemicals.

Author Response

Response to Review 3

Comments and Suggestions for Authors

The objectives of the presented study was to aimed to purify and identify the active compounds from Citrullus colocynthis leaves in order to test their efficacy against B. brassicae. The study seems to provide some interesting data on the topic. The study can be interesting for the Journal after revisions. Below I provide some indicative examples for the revision.

Comments: Abstract

Point 1: L20: TLC: give also in full

Response 1: Full detail has been given as Thin Layer Chromatography (TLC)                           

Point 2: L18: via – it is latin. Should be in italic. Check throughout the text.

Response 2: Instruction has been followed in whole of the manuscript.

Point 3: L22: in vitro and in vivo – they are latin. Should be in italic. Check throughout the text.

Response 3: Instruction has been followed in whole of the manuscript.

Point 3: L23: viz. ??

Response 3: An abbreviation for videlicet mean namely. However it was changed to “such as”

Introduction

Point 5:  Provide more convincing information about that the compounds that you purified were not previously tested against B. brassicae.

Response 5:

Point 6: L38: ‘cabbage aphid’ and not ‘Cabbage aphid’. Check throughout the text.

Response 6: Changed as cabbage aphid instead of Cabbage aphid throughout the text

Point 7: L58 and L261: Beginning of the sentences. Do not start with abbreviation. Citrullus …

Response 7: Instructions has been followed

Results

Point 8: L75:` Delete ‘Toxicity results at 50 µg/mL concentration at an exposure of 6, 12, 24, 48, 72 and 96 h are presented in (Figure 1). Interestingly,’.

Response 8: This statement has been deleted from the part 2.1 and elaborated more.

Point 9: L78: … respectively (Figure 1).

Response 9: Instruction has been followed.

Point 10: Figure 1: Give full explanation for E. Acetate; D. Water and CK.

Response 10: Full explanation for E. Acetate; D. Water and CK has been given as Acetate (Ethyl Acetate); D. Water (Distilled Water); CK (Check).

Point 11: Figure 1: Give a title for this figure. Presented title is just a footnote.

Response 11: A title for Figure 1 been given as; Figure 1. Mortality of Brevicoryne brassicae by solvent extracts of C. colocynthis leaves

Point 12: Figures 2 and 3: The titles are not well informative. Give more information.

Response 12: Titles for Figures 2 and 3 has been revised as

Figure 2a-e. The full mass spectrum for the purified compound, D3A from C. colocynthis leaves

Figure 3a-c. The full mass spectrum for the purified compound, D2 (N) from C. colocynthis leaves

Point 13: Figure 4-7 should be combined as Figure 4 A, B, C, D.

Response 13: Figure 4-7 has been combined as Figure 4a, 4b, 4c, 4d,

Point 14: Tables 2-9. Give full name of B. brassicae. df instead of D.F. Give also in full (in the footnote S.S. df, MS, and F).

Response 14: In all the tables 4-7 complete name of aphid has been written as Brevicoryne brassicae  and other footnotes in the tables are describes in full as S.S (Sum of square); df (Degree of freedom); M.S (Mean square); F (Significance); CK (Check).

Point 15: Tables 2 and 3 are the same topic. Should combined into one table.

Response 15: Combined and written as Table 2. Mortality of Brevicoryne brassicae by spinasterol, 22,23-dihydrospinasterol and fernenol via contact assay.

Point 16: Tables 4 and 5 are the same topic. Should combined into one table.

Response 16: Combined and written as Combined and written as Table 3. Mortality of Brevicoryne brassicae by spinasterol, 22,23-dihydrospinasterol and fernenol via residua assay.

Point 17: Tables 6 and 7 are the same topic. Should combine into one table.

Response 17: Combined and written as Combined and written as Table 4. Mortality of Brevicoryne brassicae by spinasterol, 22,23-dihydrospinasterol and fernenol via greenhouse assay.

Discussion

Point 18: L197:  Need more comparisons between the present results of this study and the previous studies on the compound efficacy of existing chemicals.

Response 18: Data for comparisons between the present results of this study and the previous ones has been added.

Round 2

Reviewer 1 Report

There are a couple of points to be further revised.

  1. Abbreviation in Fig. 1 is not explained in the footnote.
  2. If the extract of each ssolvent was dried and re-dissolved in each solvent for application to toxicity test, then what solvent was used for control? (line 283-285, Fig. 1 footnote)  In other words, how do the authors evaluate the toxicity of the solvent itself?
  3. In Conclusion section, provide evidence that demonstrates the compounds are safer than synthetic chemicals. Otherwise, delete "safe" in line 352.  

Author Response

Point 1. Abbreviation in Fig. 1 is not explained in the footnote.

RESPONSE 1. Abbreviation in Fig. 1 are explained in the footnote as Values in the Figure 1 are represented as mean ± standard deviation followed by different superscripts (a, b, c, d, e, f, ab, bc, cd) are not significantly different according to Duncan Multiple Range Test (DMRT) at p > 0.05. E. Acetate (Ethyl Acetate); D. Water (Distilled Water); CK (Check in distilled water); Time (h) (hours)

Point 2. If the extract of each solvent was dried and re-dissolved in each solvent for application to toxicity test, then what solvent was used for control? (line 283-285, Fig. 1 footnote)  In other words, how do the authors evaluate the toxicity of the solvent itself?

RESPONSE 2. For toxicity tests, dried extracts were dissolved in acetone (1 mL for each solvent extract) and mixed with 1% of tween-20 (prepared in distilled water) for preparing concentration. For control treatment, check (CK) was also prepared in 1% tween-20 including acetone but excluding extract to prepare 50 µg/mL concentrations

Point 3. In Conclusion section, provide evidence that demonstrates the compounds are safer than synthetic chemicals. Otherwise, delete "safe" in line 352.  

RESPONSE 3. Instructions followed and the word “safe”  has deleted.

Reviewer 2 Report

Line 44: pest but intensive and continues (should be continuous) use of

Line 99: However, D2 (N) was the mixture of two compounds (change to a mixture of two compounds)

line 99: H-NMR (600 MHz, CDCl3) δ 5.16 (m, 1H, H-7), 3.60 (m, 1H, H-3), Allylic protons….. 0.93 (d, J = 7.0 Hz, 3H, -CH3), 0.85. Check the proton spectrum.

line 115:...…. 3.60 (m, 1H, H-3) …...the allylic protons.... 1.03 (d, J = 7.0 Hz, 3H, -CH3), 0.85 (t, J = 6.8 Hz, 3H, -CH3), 0.82 check the proton spectrum

Line 121: H), 3.26–3.17 (m, 1H, -OH).....the allylic protons....1.07 (s, 3H, -CH3), 0.96 (s, 3H, -CH3 check the spectrum

line 149 and 157: (Table) remove the brackets 

206: Among the solvents extracts, methanol extract afforded

Recently, Ahmed, et al. [27] reported that solvent (what solvent) extract of C. colocynthis leaves exhibited important phytochemicals (what type of chemical) further, it possess  potential antioxidant and insecticidal activities [28]. Ahmed et al reported the identification of ……….in the solvent extract of of ……. The ethanol extract displayed potent antioxidant......

210:  chemical compounds of the plants like glycosides ( what type of glycosides), saponins, terpenoids and alkaloids. sugar moiety attached to flavonoids, saponins..... are called glycosides. 

line 295:  used for silica gal (gel) column

Author Response

Point 1. Line 44: pest but intensive and continues (should be continuous) use of

RESPONSE 1. instruction followed and the word continues changed to continuous

Point 2. Line 99: However, D2 (N) was the mixture of two compounds (change to a mixture of two compounds)

RESPONSE 2. Statement changed to a mixture of two compounds

Point 3. line 99: H-NMR (600 MHz, CDCl3) δ 5.16 (m, 1H, H-7), 3.60 (m, 1H, H-3),Allylic protons….. 0.93 (d, J = 7.0 Hz, 3H, -CH3), 0.85. Check the proton spectrum.

RESPONSE 3. Instructions followed

Point 4. line 115:...…. 3.60 (m, 1H, H-3) …...the allylic protons.... 1.03 (d, J = 7.0 Hz, 3H, -CH3), 0.85 (t, J = 6.8 Hz, 3H, -CH3), 0.82 check the proton spectrum

RESPONSE 4. Instructions followed

Point 5. Line 121: H), 3.26–3.17 (m, 1H, -OH).....the allylic protons....1.07 (s, 3H, -CH3), 0.96 (s, 3H, -CH3 check the spectrum

RESPONSE 5. Instructions followed

Point 6. line 149 and 157: (Table) remove the brackets 

RESPONSE 6. Bracket removed and the word appear as Table

Point 7. 206: Among the solvents extracts, methanol extract afforded

RESPONSE 7.

Point 8.Recently, Ahmed, et al. [27] reported that solvent (what solvent) extract of C. colocynthis leaves exhibited important phytochemicals (what type of chemical) further, it possess  potential antioxidant and insecticidal activities [28]. Ahmed et al reported the identification of ……….in the solvent extract of ……. The ethanol extract displayed potent antioxidant......

RESPONSE 8. Statements written as;

Recently, Ahmed, et al. [32] reported that solvents (methanol, ethanol, ethyl acetate, chloroform, acetone and hexane) extract of C. colocynthis leaves exhibited important phytochemicals such as alkaloids, glycosides, steroids, saponins, phenol, tannins and flavonoids along with potential antioxidant activities however, acetone and ethanol extract displayed as potent antioxidants. Further, these solvents extract also exhibited pronounced insecticidal activities [33]

Point 9. 210:  chemical compounds of the plants like glycosides ( what type of glycosides), saponins, terpenoids and alkaloids. sugar moiety attached to flavonoids, saponins..... are called glycosides. 

RESPONSE 9. . Statements is re-written as

However, the activity of the crude extract can be attributed to the existence of specific chemical compounds of the plants like fatty acids (linoleic and oleic acid) glycosides (flavonoids, phenols, saponins), terpenoids and alkaloids etc. [31]

Point 10. line 295:  used for silica gal (gel) column

RESPONSE 10. silica gal was corrected as gel column

Reviewer 3 Report

The study improved a lot. Some minor suggestions:

Introduction: Please provide more convincing information about that the compounds that you purified were not previously tested against B. brassicae.

Figure 1-2-3: In the title, please give full name of Citrullus colocynthis

L196: Discussion – without a point, (incorrect Discussions.)

L348: C. colocynthis – inside the sentence you should use short form. The full name was already introduced.

L349: B. brassicae - inside the sentence you should use short form. The full name was already introduced.

Author Response

Point 1. Introduction: Please provide more convincing information about that the compounds that you purified were not previously tested against B. brassicae.

RESPONSE 1.In the  more convincing information are added as In a study Song, et al. [22] isolated two cucurbitacins from the ethyl acetate extract of C. colocynthis fruit but were not evaluated against insects. Moreover, Ding, et al. [23] identified a mixture of spinasterol, 22,23-dihydrospinasterol from the roots of Bermeuxia thibetica but its bioactivity was not evaluated. However, Sinha, et al. [24] reported that spinasterol, 22,23-dihydrospinasterol exhibited by Melothria maderaspatana showed biological activities. It was reported that Artemisia extracts contain valuable phytochemicals which possess insecticidal activity which have been attributed mainly to the presence of fernenol and other phytoconstituents [25]. Furthermore, constituents of Artemisia vulgaris like psilostachy C, Maackiain, psilostachyin A, and fernenol possess medicinal as well as anti-bacterial activities and also used by farmers for the preservation of crops and stored grains products [26].

Point 2. Figure 1-2-3: In the title, please give full name of Citrullus colocynthis

RESPONSE 2. Instructions are followed and  a full name is written as Citrullus colocynthis in Title of the Figure 1,2,3 as

Point 3.  L196: Discussion – without a point, (incorrect Discussions.)

RESPONSE 3. Corrected as Discussion

Point 4. L348: C. colocynthis – inside the sentence you should use short form. The full name was already introduced.

RESPONSE 4. Instructions are followed Citrullus colocynthis and written as

Point 5. L349: B. brassicae - inside the sentence you should use short form. The full name was already introduced.

RESPONSE 5. Instructions are followed and written as Brevicoryne brassicae

Round 3

Reviewer 1 Report

At the first review, the authors responded as below.

Point 4: In what solvent were the extracts of different solvents, which were dried after extraction, dissolved and given to aphids?

Response 4: For toxicity tests, dried extract was dissolved in their own solvents (1 mL) by which they were extracted and to prepare concentration, 1% tween-20 prepared in distilled water was used.

At the second review, the authors' statement changed.

Point 2. If the extract of each solvent was dried and re-dissolved in each solvent for application to toxicity test, then what solvent was used for control? (line 283-285, Fig. 1 footnote)  In other words, how do the authors evaluate the toxicity of the solvent itself?

RESPONSE 2. For toxicity tests, dried extracts were dissolved in acetone (1 mL for each solvent extract) and mixed with 1% of tween-20 (prepared in distilled water) for preparing concentration. For control treatment, check (CK) was also prepared in 1% tween-20 including acetone but excluding extract to prepare 50 µg/mL concentrations

If the response of the 2nd review is truely correct, then it may be OK. I'd like to hear why this serious error has occurred from the authors.